# Wearable Sensor-Based Human Activity Recognition with Transformer Model

**DOI:** 10.3390/s22051911

**Published:** 2022-03-01

**Authors:** Iveta Dirgová Luptáková, Martin Kubovčík, Jiří Pospíchal

**Affiliations:** Department of Applied Informatics, Faculty of Natural Sciences, University of Ss. Cyril and Methodius, J. Herdu 2, 917 01 Trnava, Slovakia; iveta.dirgova@ucm.sk (I.D.L.); kubovcik1@ucm.sk (M.K.)

**Keywords:** transformer, human activity recognition, time series, sequence-to-sequence prediction

## Abstract

Computing devices that can recognize various human activities or movements can be used to assist people in healthcare, sports, or human–robot interaction. Readily available data for this purpose can be obtained from the accelerometer and the gyroscope built into everyday smartphones. Effective classification of real-time activity data is, therefore, actively pursued using various machine learning methods. In this study, the transformer model, a deep learning neural network model developed primarily for the natural language processing and vision tasks, was adapted for a time-series analysis of motion signals. The self-attention mechanism inherent in the transformer, which expresses individual dependencies between signal values within a time series, can match the performance of state-of-the-art convolutional neural networks with long short-term memory. The performance of the proposed adapted transformer method was tested on the largest available public dataset of smartphone motion sensor data covering a wide range of activities, and obtained an average identification accuracy of 99.2% as compared with 89.67% achieved on the same data by a conventional machine learning method. The results suggest the expected future relevance of the transformer model for human activity recognition.

## 1. Introduction

Human activity recognition is an important and popular research area in time series classification. Essentially, it aims at identifying human behavior based on data from sensors, available from personal devices such as smartphones, tablets, or smartwatches that can collect data from a wide sample of users and classify the signals using machine learning methods [1]. The technology of detecting human activities using mobile devices has great potential in medicine where it is possible to monitor patients with various diagnoses [2,3,4,5] and control compliance with treatment procedures or to use it as prevention against performing prohibited activities [6,7,8,9]. In addition to health monitoring and rehabilitation, this technology can be used in gaming [10], human–robot interaction and robotics [11,12], and sports [13].

A lot of effort has been focused on human activity recognition by deep neural networks. Several types of deep neural networks are typically used for time series classification of sensor signals, such as convolutional neural networks [14,15,16], fully convolutional neural networks [17], multiscale convolutional neural networks [18], time-LeNet [19], stacked denoising autoencoder [20], deep belief neural networks [21], Long Short-Term Memory (LSTM) deep recurrent neural network [22], echo state networks [23], or inception-ResNet [24]. 

Among the many previous studies on the application of deep learning on smartphone motion sensor data, several studies have been selected here to be discussed more deeply and compared to the approach proposed in the present study. Zebin et al. [22] applied exclusively LSTM layers with the addition of Batch Normalization layers. In this configuration, the time series could only be traversed in one direction, from zero to the maximum time step. However, the transformer model in the present study allows one to traverse the time series arbitrarily, since all the contextual relationships are calculated in parallel. When the attention mask is defined, the transformer model can search for connections among the features in the time series both in the direction from past to future, as well as in the opposite direction from future to past, according to the trained attention matrix.

Quin et al. [25] transformed the measured signal time series into a polar coordinate system, and formed a pair of Gramian Angular Field images. Then, these images were classified by a ResNet-based convolutional deep neural network. Compared to their approach, the present study focuses on the direct application of measured signals to the input of a neural network, eliminating the need for any complex pre-transformation of data. Moreover, the model in this study is much smaller than ResNet used in [25]. The overall transformation and subsequent classification are ensured within the trained model, and therefore, it is possible to achieve higher speeds during prediction and it is not necessary to use demanding calculation models such as ResNet. Normalization is ensured, therefore, the mean is close to zero and the standard deviation is 1.0 as compared with the min–max method used by Qin et al. [25].

Wang et al. [26] primarily focused on the application of one-dimensional (1D) CNN and LSTM. This combination could handle long time series, but it was not as effective as the transformer model in massive parallelization of calculations. The transformer itself can process long time series at high speed without the need to combine multiple neural network approaches. Although the dataset used in [26] contained several times more examples than the dataset used here, it focused only on basic activities and transition activities, not on more complex movements such as pick, jump, push-up, or sit-up used in the present study. Moreover, in this study, a data augmentation method is introduced for extending a dataset by manipulating existing data, making it possible to produce many new examples to supplement an existing dataset.

Gao et al. [27] introduced a new approach—attention for learning multiscale features among multiple kernels of 1D convolution layers in HAR issues. In a similar way, the signals were preprocessed to zero mean and one standard deviation, but their focus was on using special 1D convolution layers for the prediction of one label for the entire time series (window). One-dimensional convolutional networks and recurrent neural networks, or a combination of both are among the most used approaches, see [28]. In the present study, the window size limit is restricted only by the memory capacity, and labels are assigned to each time step, using the previous steps in the time series. This study also offers an alternative in signal processing to that of 1D convolution in the form of entirely fully connected layers.

Li et al. [29] joined 2D convolution layers followed by a BiLSTM layer. The BiLSTM allows the time series to move in two directions, from the past into the future and from the future into the past. However, unlike the transformer model, it works on the principle of two LSTM layers within one, when the other layer is presented with a reverse copy of the original time series. When merging this pair of layers into one output of the bidirectional layer, it is possible to use the following methods: Sum, multiple, concat, and average [30]. With the transformer model, this is a natural feature and there is no need to make reverse copies or look for the most convenient method of merging LSTM pairs of bidirectional layers within the hyperparameters.

Gupta [31] described the classification of simple as well as complex activities by widely used models such as InceptionTime or DeepConvLSTM. The activities were captured using sensors in smartphones, similar to the present study, and smartwatches. Combined convolutional and recurrent neural networks were used for evaluation. Gholamiangonabadi et al. [32] focused on comparing feed-forward neural networks and convolutional neural networks in terms of cross-validation on unseen subjects. The present study offers an alternative that directly focuses on using attention mechanisms to find connections in the time series between features. 

Alemayoh et al. [16] converted the features in a time series into a grayscale image with shades of gray expressing the measured value from the sensor. The merging of data from all axes of the accelerometer and the gyroscope produced a 2D image, which was processed by a 2D convolutional neural network. Its input could be arranged into one common frame or, when each sensor was processed separately, the frame was specified by a channel (similarly to the RGB frame, where there were three channels). In the case of the J48 decision tree method, Fourier and wavelet transform preprocessing as well as feature acquisition were used. This allowed the signal to be divided into portions of high frequencies representing noise and low frequencies representing real activities. In contrast, the present study focuses on finding advantageous alternatives to classical approaches such as convolutional neural networks or recurrent neural networks. Using attention blocks, the transformer directly focuses on predicting the intensity of the gain/loss of the feature during the feed-forward phase based on the context found during the learning process. Two-dimensional convolutional networks work directly on the principle of the image classifier, and thus learn to recognize hidden patterns in signal frames and predict activities from them. The transformer model is a universal architecture used similarly to convolutional neural networks in NLP, vision, or signal processing tasks.

The present study compares directly with the Sikder and Nahid method [30], as it is based on their measurements. Each activity was recorded in a 300-time step window width with a sampling frequency of 100 Hz, which corresponded to 3 s of human activity. However, the data in [33] required complex preprocessing and extraction of significant features, which allowed methods such as random forest to classify activities relatively accurately. When preprocessing signals, it is possible to use fast Fourier transform, which extracts frequency-domain features from the input signal and at the same time suppresses, to some extent, the effect of noise on the classification [33]. 

The current manuscript deals with the application of deep neural networks directly to the normalized time series of the signal from the sensors. This study employs an alternative approach to processing time series based purely on the attention mechanisms, called transformer. The transformer model directly focuses on using attention mechanisms to find correlations in the time series between features and allows massive parallelization of time series calculations, which is different as compared with recurrent neural networks that iterate serially through a time series. Another advantage of the transformer is the longer path length between features in the time series, which allows for more accurate learning of the context in long time series, an assertion stated by Vaswani et al. [34]. Computation speed, as well as prediction accuracy, are key elements in working with human activities, where prediction can be performed directly on the mobile device. The sequence-to-sequence method is used in the prediction of activities [35], where all time steps from the transformer output are considered and activity designations are assigned to them. In this way, it is possible to assign activity to each time step that the user has taken when measuring live values from a mobile device.

A comparison of the selected methods, data preprocessing, datasets, and accuracy for wearable sensor-based human activity recognition from seminal and state-of-the-art papers is provided in Table 1.

The primary contribution of the present study is a better deep learning model with an attention mechanism for activity recognition from mobile data. The model itself is not our invention, but we proposed its adaptation for application in human activity recognition, which is outside its typical scope of use. The results fully support the proposed suitability of the transformer model for human activity recognition. 

The remainder of this paper is organized as follows: In Section 2, we present the details of the general transformer model, the vision transformer model, the used KU-HAR database of smartphone motion sensor data, our adaptation of the transformer model for human activity recognition, data augmentation of the KU-HAR dataset, and optimization of the metaparameters of our model; the results and their analysis are described in Section 3; in Section 4, we discuss the results and compare them with the state-of-the-art methods; we draw conclusions and outline directions for future research in Section 5.

## 2. Methods

In this section, we describe the deep neural network method, the transformer model, and its extension, and the vision transformer model, which form the foundation for the method described here. Then, the details about the used KU-HAR dataset are further provided. Adaptation of the transformer method designed, here, for human activity recognition is followed by augmentation of the used dataset for combined couples of activities. The analysis of the hyperparameters of the proposed transformer method adaptation leads to their optimization.

### 2.1. Transformer Model

A transformer model is a deep learning neural network, where the attention mechanism provides context for any position in the time series. Similar to recurrent or convolutional networks [41], transformers efficiently handle time series classification and employ correlations between features within time steps. They are frequently used to work with natural language, where they achieve higher scores than recurrent neural networks. Before the transformer model was introduced, the cutting-edge systems for complex time series problems such as natural language processing were based on gated recurrent neural networks with a superimposed attention mechanism [42]. The transformer model shows that the attention mechanism does not need to use recurrent units to achieve equivalent performance. Moreover, the transformer model is much easier to parallelize than convolutional neural networks. 

In the present study, the principle of the transformer model was not modified as compared with that in [34], only its input and output were changed. The transformer model consists of multi-head attention layers, fully connected layers, normalization layers, and dropout layers [34]. It also contains residual connections that help with the gradient backpropagation in a deep neural network. 

The attention mechanism is crucial to the transformer model, where each attention head in multiple attention heads can search for a different definition of relevance or a different correlation. Multi-head attention [34] is based on the principle of mapping a query and a set of key-value pairs to an output. The output of the network is the weighted sum of values, where the weight is assigned to each value (V) based on the calculation of the compatibility function from the query (Q) and the corresponding key (K). Dot products of the query and all keys are calculated, and then the softmax function is applied to normalize the obtained weights, which are multiplied by values, see Equation (1). Softmax is the basis for categorical distribution prediction, where its output values define the probabilities of individual categories. Normalization comes in the sense of conversion from the interval [-infinity, infinity] to the interval [0, 1]. Values in the interval [0, 1] can be directly understood as some volume intensity of the features (comparable in a sense gain/mute) obtained in a given time step. With softmax, it is not true that only one output has a high probability and the other outputs have low probability. In the present model, it is a continuous distribution of intensities of different values, where, after matrix multiplication with values, these intensities determine what information is to be further propagated by the neural network.

Multi-head attention contains several modules called heads that have their own queries and a set of key value pairs expressed by fully connected layers from the original queries and a set of key value pairs fed to the input layer, see Equation (2). The advantage of using multiple heads lies in the ability to combine the different contexts found from each of the heads into one complex output.

The outputs from each head of multi-head attention layers are combined into one fully connected output layer, see Equation (3). This layer is followed by the position-wise feed-forward network block, which is composed of a pair of fully connected layers linked by the nonlinear activation function RELU (rectified linear unit), see Equation (4). Typically, the number of neurons in the first fully connected layer is 4 times higher than that in the following layer, where the number of neurons is equal to the number of features entering this block. The entry determining the position of the features within the series is also added to the network input, because the transformer does not know the order of the features, for example, within the processed sentence [34]:(1)$$Attention\left(Q,K,V\right)=softmax\left(\frac{QK^{T}}{\sqrt{d_{k}}}\right)V$$
(2)$$head_{i}=Attention\left(QW_{i}^{Q},\;KW_{i}^{K},VW_{i}^{V}\right)$$
(3)$$MultiHead\left(Q,K,V\right)=Concat\left(head_{1},\;\ldots ,\;head_{h}\right)W^{O}$$
(4)$$FFN\left(x\right)=\max\left(0,xW_{1}+b_{1}\right)W_{2}+b_{2}$$

### 2.2. Vision Transformer Model

For image classification tasks, the transformer model replaces the typically used convolutional networks with satisfactory results. The image is divided into patches forming a sequence of features to which attention mechanisms are applied. However, a very large dataset of images of 14–300 million examples is needed for the transformer model to achieve excellent results. The advantage of using a transformer in image classification tasks is its speed and scalability. In contrast to the original transformer encoder [34], the normalization layer is applied before each block and residual connections after each block. The nonlinear activation function used in the position-wise feed-forward network block is the Gaussian error linear unit. The position of individual patches within the overall processed image is determined by the parameters that are adapted in the learning process together with the neural network [43].

### 2.3. KU-HAR Dataset

The KU-HAR dataset used in the current manuscript was published by Sikder and Nahid [33]. It was chosen here among other data sources, because it contains a lot of examples divided into 18 classes (activities). The human activity recognition (HAR) data were obtained from 90 participants aged 18 to 34 years. The ratio of women to men among the participants was 1:5. The weight range of the participants was from 42.2 to 100.1 kg. The dataset contains 20,750 preprocessed examples, where each example captures 3 s of the performed activity, i.e., one whole time series of the signal represents just one performed activity and has only one label assigned to it. The measurements used sensors in a smartphone (accelerometer and gyroscope [33]) placed in a waist bag on each participant. The smartphone was facing the left side in the bag and the screen was pointing in the same direction as the participant. The first 11 activities presented in Table 2 were recorded inside a classroom because they did not need a large space to perform. The other 4 activities, described next, were recorded outside the classroom, but within the university building. The run activity was recorded in the corridor in front of the classroom, stair-up and stair-down activities were recorded on the staircase between the ground floor and the third floor, where there were 3 staircases between each floor. The table-tennis activity was recorded in the common room located on the ground floor. 

The preprocessing consisted of deleting the part of the data recorded before the start of the performed activity because the first seconds of the recording did not correspond to the actual start of the performed activity. Similarly, the unrelated part of a record was removed at the end of an activity scan. 

The next step of the preprocessing was to unify the sampling frequency from all measurements to 100 Hz. Because different smartphones with different computing powers were used, not all measurements were identical concerning sampling frequency, and therefore, one-dimensional interpolation of recorded time data was used when a particular measurement was recorded. The last step was to divide the measured activities into time series with a fixed length of 3 s. Each time series contains a unique portion of the original measurements [33].

### 2.4. Transformer Model for Human Activity Recognition

The examples of the vision transformer model have shown how effectively transformer models can replace existing recurrent and convolutional neural networks. The vision transformer model works with signals in the form of an image, which supports an assumption that it can also process 1D time series of signals from sensors such as an accelerometer or gyroscope. The transformer model for human activity recognition presented further is based directly on the vision transformer model architecture [43], where, however, the signal is fed directly as input into the encoder block along with the added information determining the position of the features within the signal time series. 

The transformer model for human activity recognition operates in a sequence-to-sequence mode and predicts the class for each time series feature, see Figure 1. The advantage is that if there are several consecutive classes in one time series, these classes can be easily identified, and the transformer is not limited to the features in the whole time series belonging to one class. All fully connected layers are initialized using the truncated normal distribution with a standard deviation of 0.02, as in BEIT [44]. Before the signal is fed as an input to the neural network, it passes through a normalization layer that stores the mean and variance obtained from the training data and adjusts the input to the values of 0.0 mean and 1.0 standard deviation. The advantage of this solution is that when the model is put into practical use, it already contains these calibration values, and it is not necessary to solve the signal adjustment in an external way. The model is completely ready for implementation in mobile devices, provided that the measured quantities are in the basic physical units, the accelerometer in m/s^2^, and the gyroscope in rad/s. The output layer of the model is linear, to provide a higher computational speed on less powerful devices than if the softmax function was used. In principle, the maximum value corresponding to the predicted activity can also be obtained before the application of softmax, which is only used during learning in the loss function. This form is used there, due to the calculation with logarithms, since it does not have negative numbers at the output, as the linear layer does. 

The principal task in working with the signal time series is to find the correlations in it that will most positively affect the classification result. From the time series, the features are mutually matched by attention mechanisms, and therefore, this transformer model ranks among the self-attention mechanisms, the same signal is applied to the input query, key, and value [33]. The advantage of using a transformer in signal processing is again its speed and scalability, which has an impact on the usability of mobile devices and the accuracy of class predictions. The experiments show that it is a suitable alternative to recurrent and 1D convolutional networks for signal classification tasks. As with vision transformers, a huge number of examples are required, therefore, in this study a special data augmentation method has been devised for signals expressing various human activities.

The implementation of the transformer model for human activity recognition (see Appendix A) was realized using the open-source library TensorFlow [45], which contained a rich set of tools for neural network design, as well as its learning, evaluation, and deployment. It contained all the basic layers for transformer model creation: normalization layers, multi-head attention layer, dropout layer, fully connected (Dense) layer, up to the position embedding layer that was created as a custom layer by inheriting from the basic class layer. The encoder block was also created as an advanced custom layer from several simpler layers for easy replication. TensorFlow was also used here for its professional deployment in many international companies and its high performance in mobile and embedded devices in the form of TensorFlow Lite.

### 2.5. Data Augmentation

To extend the KU-HAR dataset so that more training examples were available, an algorithm was chosen to combine pairs of activities that could follow each other in real life with a high probability. It was necessary to create all combinations of activity pairs from the original dataset, with the provision that identical activities were excluded. These resulting pairs had to be manually checked and their logical sense verified, see Table 3. The next step was to combine these activities into a double-length window, which needed to be transformed into a standard number of time steps used for previous training samples. The downsampling method was chosen, omitting every second step from the time series [46]. Vision transformers were taught in a similar way, where randomly selected parts of the image were replaced by noise, and the transformer model aimed to fill these places identically to the original image [47]. For sensor data, here, it was not necessary to replace the omitted time steps with noise, but the network must also process the signal and correctly identify the performed activity. As can be seen in Figure 2, the numbers of examples in the classes substantially differ, and therefore, it was necessary to choose the method by which the examples of activity pairs would be generated. The smaller of number of examples of both classes in pairs of activities was used. This avoids duplication of examples of the paired activity with fewer examples, which could cause overfit [48]. The transformer model also acquired a logical awareness of the connections between possible successive actions, which would not be possible with the coupling of pure random pairs of activities. In total, 83,129 examples were obtained from the original 20,750 examples. Then, the dataset was divided into training, testing, and validation sets in a ratio of 70:15:15%. Manipulation with the dataset was carried out by NumPy libraries [49] for working with matrices, Pandas [50] for working with CSV files, and Scikit-learn [51] for even distribution of examples according to classes per training, testing, and validation datasets.

The distribution of examples by class can be seen in Figure 2. It is apparent, that the used dataset is slightly unbalanced. “Final” represents the distribution of examples after applying data augmentation, ”New” represents newly created examples from a combination of original examples, and “Original” is a distribution of examples from the original dataset.

The lay-stand and stand-sit classes were omitted from pair combinations, as they represented a transition activity already composed of a pair of activities. The goal was to combine just two different activities and a combination of lay-stand or stand-sit with another class would create windows with up to three activities. Another completely omitted activity was push-up, there was no suitable activity to pair it with, which would occur immediately before or after this activity. The only logically close activity was lay, but it was “performed” on the back.

### 2.6. Finding Optimal Hyperparameters

The hyperparameters were optimized by the WanDB Sweep tool [52], which not only provides parallel coordinates chart for visualization of various settings but also offers a prediction of importance and correlation of hyperparameters against the selected metric. The used metric was the best validation accuracy obtained from the best prediction over the validation dataset during the learning process. The search method was random, which selected settings from the predefined ranges of the hyperparameters. Progressively, these intervals were manually adjusted to increase the accuracy of the model, and finally, the most suitable combination of them was chosen, see Table 4.

The level of importance of the individual hyperparameters for maximizing the best validation accuracy metric is depicted in Figure 3. The way in which they affect this metric is denoted by color. The red color indicates a negative correlation and the green color a positive correlation.

From Figure 4, according to the color scale, it is possible to determine how the given value of the hyperparameter influenced the best accuracy in the predictions on the validation dataset. Yellow expresses the highest accuracy in predictions and blue the lowest.

The combinations of hyperparameter values plotted on multiple y-axes is shown in the Figure 4. The last part of the chart consists of a color scale, with light yellow color presenting the highest accuracy and dark blue color the lowest accuracy. From the graph, it is possible to deduce the most suitable intervals of values of hyperparameters, and thus, guide the search for the most advantageous combinations. This technique is advantageous for a guided random search used in this study when one can choose from many possible combinations generated from predefined distributions [52]. Such tuning of hyperparameters has an advantage both over the Bayesian hyperparameter optimization as well as over the grid search, in this study, for these methods there are too many hyperparameters to tune optimally. 

## 3. Results

In the testing phase, previously unseen examples from test datasets were presented to the neural network. The task was to predict activities from hitherto unseen signals with the highest possible accuracy. Figure 5 shows the individual attention matrices from Head 1 of the transformer model expressing only one activity.

The uppermost row of panes in Figure 5 shows the attention matrices during static activities when the participants did not perform any movement. Example activities stand, sit and lay were selected for this purpose. The second row shows the attention matrix for dynamic activities, where participants performed active movements. The table-tennis, walk, and jump activities were chosen for this purpose. Attention heatmaps show the intensity of the relationship between features in different time steps, the paler the color of a pair of features from different time series, the higher the intensity of the relationship. The black color expresses pairs without any significance of their link for feed-forward and successful activity classification. 

In the second row of Figure 5, it can be noticed that the model is so influenced by the used data augmentation method that it tends to divide the time series into two halves, even if it is one activity. This may indicate a problem with the transformer model when it does not have enough examples and, thus, does not properly generalize to different variations of activity ratios within the time series. Only a 1:1 ratio was used in this paper.

Similarly, Figure 6 shows selected examples of attention matrices from Head 1 of the transformer model expressing activity pairs. From the attention matrices, you can see the transition of activities in the exact half of the time series, in which the first half belongs to one activity and the second half belongs to another activity. This proves the effectiveness of the transformer algorithm in matching features from time series that contribute to correct classifications.

To compare the differences between paired static and dynamic activities, the following pairs were chosen: ”stand/stair-down” represents a static to a dynamic connection, ”sit-up/lay” represents a dynamic to a static connection, and ”pick/table-tennis” represents a dynamic to a dynamic connection.

The cosine similarities of the positions of features within the time series can be seen in Figure 7. This visualization is made possible by learning position tags in the form of position embedding. As a result, the transformer model customizes the feature label within the time series. Therefore, it is possible to see from Figure 7 which positions are similar. It is also possible to see the division into four square parts, which symbolize the similarity within the first and the second 150-time step window. This divides the whole time series of 300 steps in half. This half-and-half division is caused by the implemented method of data augmentation, where pairs of activities are combined only in a 1:1 ratio, one after another. However, when using other ratios, there is a higher probability of losing essential information from the signal during downsampling; currently, there is a loss of half of the information from the signal from both parts of the time series.

The colors expressing similarity from pink to the dark red scale indicate positive similarity between the vectors expressing the positions of the features within the signal; the lighter the pink is, the more these positions are similar. Colors in the blue scale indicate negative similarity of positions; light blue indicates dissimilar (opposite) vectors.

Figure 5, Figure 6 and Figure 7 visualize the resulting knowledge gained during the training process and Figure 8 shows the exact opposite, i.e., a random model without any knowledge before the learning process. 

## 4. Discussion

The adapted transformer model proposed in this manuscript for human activity recognition was studied with the goal to demonstrate its suitability as a viable alternative to various versions of recurrent or convolutional neural networks. This goal was supported by findings, that in other applications, the transformer model often seems to have better representation power than the otherwise popular long short-term memory architecture of an artificial recurrent neural network (RNN) [53]. The transformer can scale up the model to more than one million parameters and can also be used on mobile devices. It can push the measured signal time series directly (after a normalization) into the neural network, without the need for pre-transformation of the data. Moreover, the transformer is also well parallelized to run on GPU [33].

The quality of the result can be seen in Figure 9. The biggest numbers in the confusion matrix of Figure 9a are on the diagonal, which is also indicated by dark blue. This shows that the predicted labels mostly match the true labels. Figure 9b shows another statistical evaluation of the results. Values of precision, recall, f-1 score, and support are presented for each of the activities. The support, i.e., the numbers of samples of the true response that lie in the activity, range roughly from 2 × 10^4^ to 44 × 10^4^, which are reasonably large support sets. The precision ranges from 0.945 for the activity talk-sit, to 1.0 for the table-tennis activity. While it seems, that table-tennis activities cannot be mistaken, from the confusion matrix it is apparent, that the talk-sit activity is most often confused with sit activity. Apparently, talking probably does not show significantly either on the accelerometer or gyroscope data.

The proposed HAR transformer model achieved, on average, 99.2% prediction success as compared with the original 89.67% of the KU-HAR work by [33]. It successfully coped with the classification of one activity contained in the whole time series as well as with the merging of two activities in one time series. The robustness of the predictions was not affected by the omission of every second signal measurement from the time series. A new method of signal data augmentation has also been devised, focusing on the logical connections between signals and their appropriate impact to enhance the accuracy of the transformer model predictions. The results of the experiments showed how the attention mechanisms found correlations in the long time series of the signal and further promoted the most important of them, which positively affected the classifications of activities. 

## 5. Conclusions

The results presented in this manuscript showed the benefits and utility of the transformer model in classifying human activities. The dataset selected for testing is the largest currently available for smartphone motion sensor data, covering a wide range of activities [33,36,37,38,39,40]. However, since it was published in 2021, it has not yet been used for HAR prediction by state-of-the-art hybrid convolutional neural networks with either bidirectional long short-term memory or other deep learning models, which are considered among top contenders [54]. The random forest method prediction accuracy on this dataset [36] was slightly lower than deep learning methods achieved on other datasets. However, the adapted transformer model proposed in the present study for HAR achieved a level of precision that suggest it has a potential to be included among cutting-edge methods for HAR, see Table 1 in Section 1. 

In the future, the adapted transformer model should be tested on an enlarged dataset, ideally using different sensor data. Then, the results should be usefully applied, at first for models of robots, which should serve as a springboard for further useful applications involving direct support for humans.

## Figures and Tables

**Figure 1 sensors-22-01911-f001:**
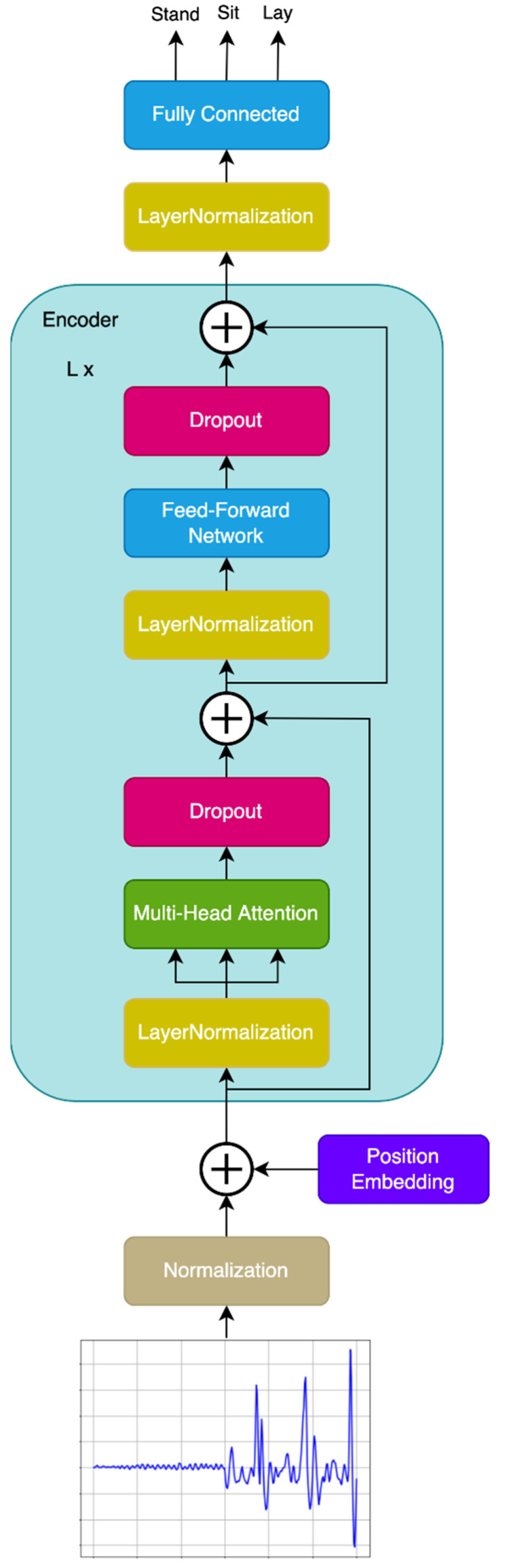
The transformer model for human activity recognition.

**Figure 2 sensors-22-01911-f002:**
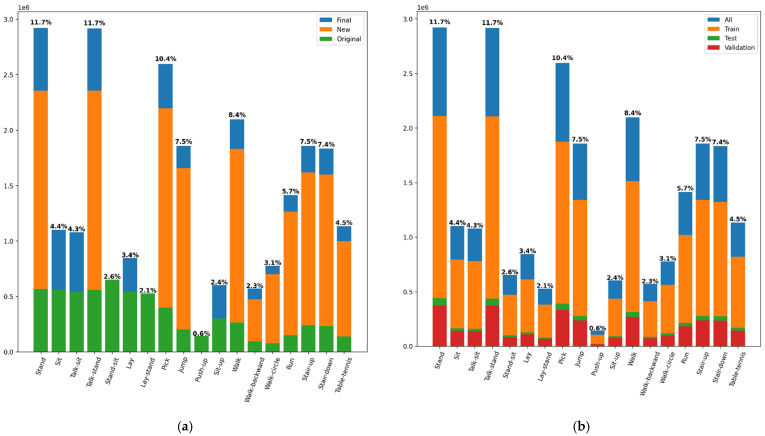
Distribution of examples by classes: (**a**) Distribution of examples by individual classes before and after the data augmentation process; (**b**) distribution of examples by individual classes into training, testing, and validation datasets.

**Figure 3 sensors-22-01911-f003:**
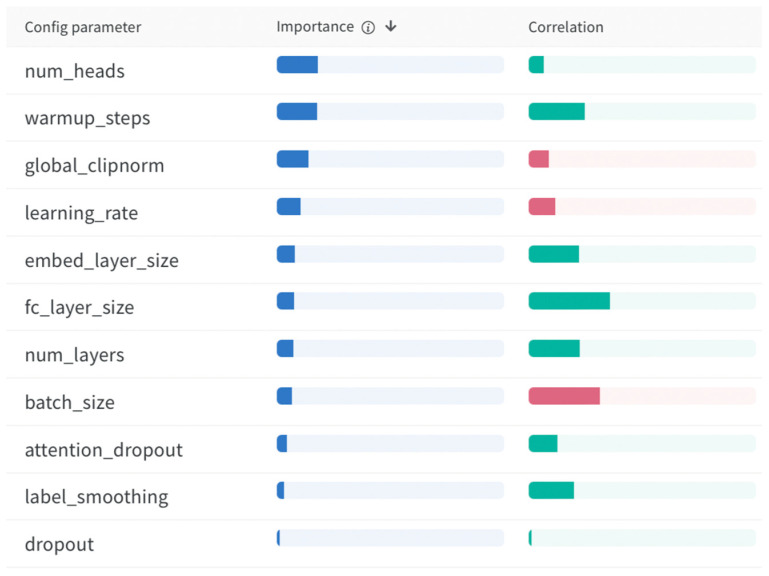
Parameter importance chart.

**Figure 4 sensors-22-01911-f004:**
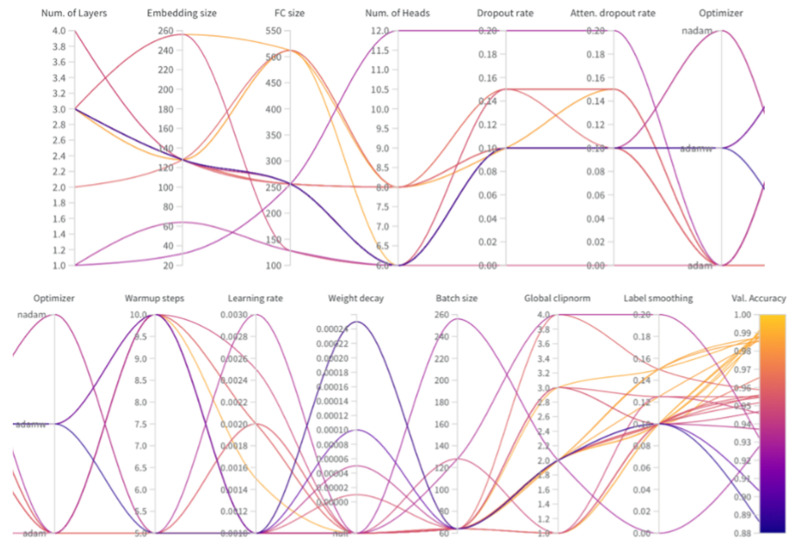
Parallel coordinates chart, split in half into two rows.

**Figure 5 sensors-22-01911-f005:**
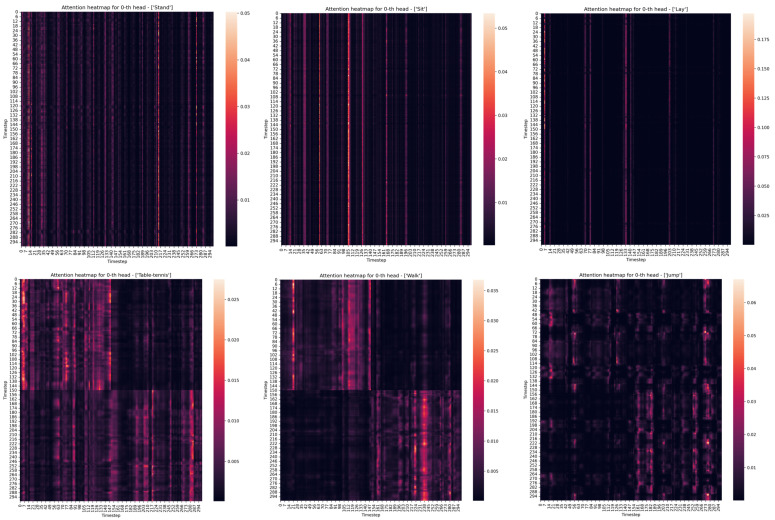
Attention heatmaps of different single activities.

**Figure 6 sensors-22-01911-f006:**
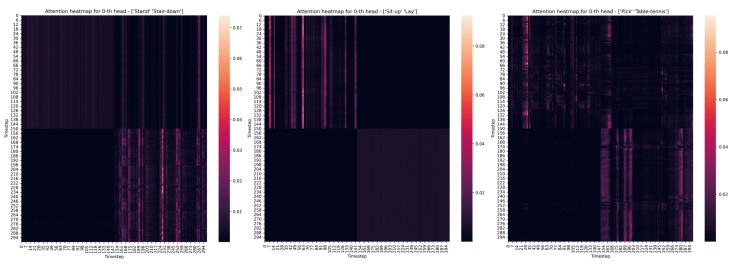
Attention heatmaps of different pairs of activities.

**Figure 7 sensors-22-01911-f007:**
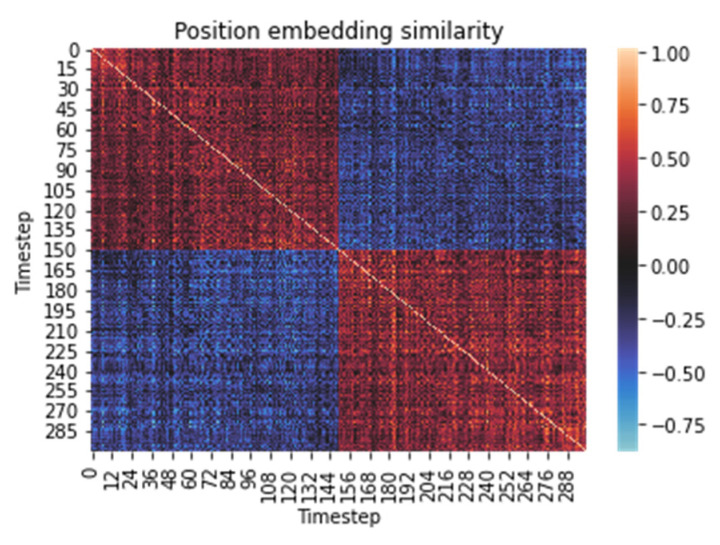
The cosine similarity between the position embedding of the time step.

**Figure 8 sensors-22-01911-f008:**
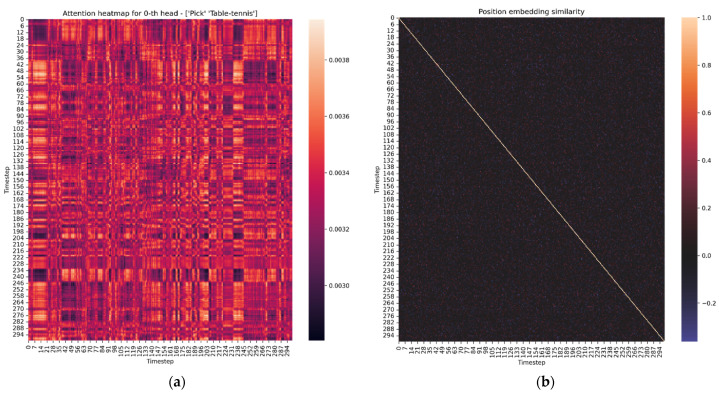
Visualization of: (**a**) Attention matrix; (**b**) position embedding similarity of randomly initialized transformer.

**Figure 9 sensors-22-01911-f009:**
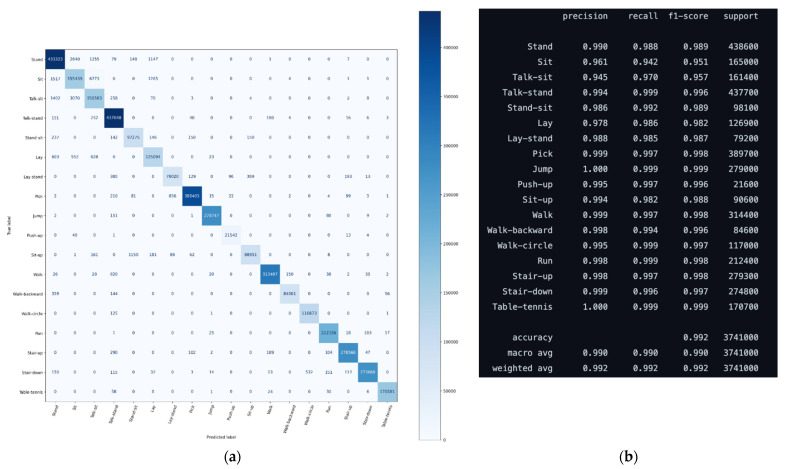
(**a**) Confusion matrix; (**b**) class-wise performance of the transformer model for human activity recognition.

**Table 1 sensors-22-01911-t001:** Comparison of selected methods, preprocessing, and resulting accuracy over different datasets for HAR based on mobile sensors data.

Standard Dataset	Paper	Data Structure	Method	Accuracy
KU-HAR [33]	This study	Standardization	HAR transformer	99.2
	Sikder et al. [33]	fast Fourier transform	Random forest	89.67
MHEALTH [36]	Qin et al. [25]	Gramian angular fields	GAF and ResNet	98.5
PAMAP2 [37]	Li et al. [29]	Standardization	2D Conv + BiLSTM	97.15
	Gao et al. [27]	Standardization	Conv + SKConv	93.03
WISDM [38]	Alemayoh et al. [16]	Segmentation into a grayscale image that represents the time serie of signal	SC-CNN	97.08
	Gupta [29]	RAW	CNN-GRU	96.54
	Alemayoh et al. [16]	Heuristic features	J48 decision tree	90.04
HAPT [39]	Wang et al. [26]	Splice into two-dimensional matrix (like a picture)	CNN-LSTM	95.87
UK Bio-bank [40]	Zebin et al. [22]	RAW	LSTM + BN	92

**Table 2 sensors-22-01911-t002:** Description of the activity classes in the KU-HAR dataset, amended from [33].

Class Name	ID	Performed Activity	Duration Repetitions	No. Subsamples
Stand	0	Standing still on the floor	1 min	1886
Sit	1	Sitting still on a chair	1 min	1874
Talk-sit	2	Talking with hand movements while sitting on a chair	1 min	1797
Talk-stand	3	Talking with hand movements while standing up or sometimes walking around within a small area	1 min	1866
Stand-sit	4	Repeatedly standing up and sitting down (transition activity)	5 times	2178
Lay	5	Laying still on a plain surface (a table)	1 min	1813
Lay-stand	6	Repeatedly standing up and laying down (transition activity)	5 times	1762
Pick	7	Picking up an object from the floor by bending down	10 times	1333
Jump	8	Jumping repeatedly on a spot	10 times	666
Push-up	9	Performing full push-ups with a wide-hand position	5 times	480
Sit-up	10	Performing sit-ups with straight legs on a plain surface	5 times	1005
Walk	11	Walking 20 m at a normal pace	~12 s	882
Walk-backward	12	Walking backwards for 20 m at a normal pace	~20 s	317
Walk-circle	13	Walking at a normal pace along a circular path	~20 s	259
Run	14	Running 20 m at a high speed	~7 s	595
Stair-up	15	Ascending on a set of stairs at a normal pace	~1 min	798
Stair-down	16	Descending from a set of stairs at a normal pace	~50 s	781
Table-tennis	17	Playing table tennis	1 min	458
			**Total**	20,750

**Table 3 sensors-22-01911-t003:** Newly created couples of activities.

Stand + Talk-Stand	Sit + Talk-Sit	Talk-Stand + Stand	Pick + Stand	Jump + Stand	Walk + Stand	Walk-Backward + Stand	Walk-Circle + Stand	Run + Stand	Stair-up + Stand	Stair-down + Stand	Table-Tennis + Stand
Stand + Pick	Talk-sit + sit	Talk-Stand + Pick	Pick + Talk-Stand	Jump + Talk-Stand	Walk + Talk-Stand	Walk-backward + Talk-Stand	Walk-circle + Talk-Stand	Run + Talk-Stand	Stair-up + Talk-Stand	Stair-down + Talk-Stand	Table-tennis + Talk-Stand
Stand + Jump	Lay + Sit-up	Talk-Stand + Jump	Pick + Jump	Jump + Pick	Walk + Pick	Walk-backward + Pick	Walk-circle + Pick	Run + Pick	Stair-up + Pick	Stair-down + Pick	Table-tennis + Pick
Stand + Walk	Sit-up + Lay	Talk-Stand + Walk	Pick + Walk	Jump + Walk	Walk + Jump	Walk-backward + Jump	Walk-circle + Jump	Run + Jump	Stair-up + Jump	Stair-down + Jump	Table-tennis + Jump
Stand + Walk-backward		Talk-Stand + Walk-backward	Pick + Walk-backward	Jump + Walk-backward	Walk + Walk-circle	Walk-backward + Table-tennis	Walk-circle + Walk	Run + Walk	Stair-up + Walk	Stair-down + Walk	Table-tennis + Walk
Stand + Walk-circle		Talk-Stand + Walk-circle	Pick + Walk-circle	Jump + Walk-circle	Walk + Run		Walk-circle + Run	Run + Walk-circle	Stair-up + Walk-circle	Stair-down + Walk-circle	Table-tennis + Walk-backward
Stand + Run		Talk-Stand + Run	Pick + Run	Jump + Run	Walk + Stair-up		Walk-circle + Stair-up	Run + Stair-up	Stair-up + Run	Stair-down + Run	Table-tennis + Walk-circle
Stand + Stair-up		Talk-Stand + Stair-up	Pick + Stair-up	Jump + Stair-up	Walk + Stair-down		Walk-circle + Stair-down	Run + Stair-down	Stair-up + Stair-down	Stair-down + Stair-up	Table-tennis + Run
Stand + Stair-down		Talk-Stand + Stair-down	Pick + Stair-down	Jump + Stair-down	Walk + Table-tennis		Walk-circle + Table-tennis	Run + Table-tennis			
Stand + Table-tennis		Talk-Stand + Table-tennis	Pick + Table-tennis	Jump + Table-tennis							

**Table 4 sensors-22-01911-t004:** Optimized hyperparameter settings.

Name	Description	Value
Epochs	Number of training episodes	50
Attention dropout rate	Dropout applied to the attention matrix	0.1
Batch size	Number of samples applied during training at once	64
Dropout rate	Dropout applied between layers	0.1
Embedding size	Size of features after projection signal and size of position embedding	128
Fully Connected (FC) size	Size of the first layer in the position-wise feed-forward network	256
Global clipnorm	Clipping applied globally on gradients	3.0
Label smoothing	Smoothing of the hard one-hot encoded classes	0.1
Optimizer	Optimizer used during training model	Adam
Warmup steps	Number of steps from the learning starts to reach learning rate maximum	10
Learning rate	The maximum value of learning rate after warmup	0.001
Learning rate scheduler	The scheduler that controls the learning rate during training	Cosine
No. Heads	Number of heads in multi-head attention	6
No. Layers	Number of encoder blocks in the entire model	3

## Data Availability

Not applicable.

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
