# Peer review of "Wearable Sensor-Based Human Activity Recognition with Transformer Model"

_sensors, 2022, doi:10.3390/s22051911_

Round 1

Reviewer 1 Report

The present work describes a Wearable Sensor Based Human Activity Recognition with Transformer Model

My advises are below, after deep review of it:

---------------

Title

---------------

I recommend to change the Title to: "Wearable Sensor Based Human Activity Recognition with Transformer Model".  

The word Transformer alone, sounds strange, since this is a model.

---------------

Abstract

---------------

L12: Please, remove word "successful". This kind of words aren't  recommended to use, since it looks like you are  talking about yourself. It is self praise. The success or not of some application must to be said for the readers/reviews, public, not for the author.

L14-15: Remove sentence from "Transformer model is ...." until "..well on signal". It is not usual to put term definition in the abstract. 

-       In resume, I recommend to rewrite all abstract. It is not well structured and the writing isn't good. See examples of another accepted journals to get some tips.

---------------

Introduction

---------------

The introduction is badly wrote. Mainly the paper citation. It not correct writing for MDPI.

L25-27: Please, rewrite the sentence.

L25: "very active.." sounds strange in the journal context.

L31: Please put reference for each field of application you cited. 

L33: Remove "recently". Temporal word are not recommended to use in any paper.

L34: Which DNN types?? Please cite some and references. 

L34-35: Please rewrite the sentence. The author suddenly started to cite references "Paper [2]". It is not good practice. Please, cite author name on a  correct citation format.

L46: Please remove from all journal the format "Paper [REFERENCE]". It is not usual on MDPI journals. Please, see MDPI template and change these  writing formats.

L46-94: Please, I strongly recommend to rewrite all texts who say "The paper []...." It is really strange on the text, jumping to another paragraph, using "The" before "Paper[]"...and so on. Really confuse and bad wrote sentences and paragraphs.

L81: The author said.. "This paper.." Which paper? Where is the paper? It is no correct use of reference. Please, rewrite all.

---------------

Methods

---------------

L98: Change title to "Transform Model".

L100: "..typically IT process..."

L99-101: Confuse sentence. Please, rewrite it. Remove "which" and adapt the last part of the sentence.

L103: "Normalization" or "normalized"?

L106: The author SUDDENLY presents a paragraph to define Multi-Head Attention. Please, try to introduce the MHA definition in a smooth way between the text, not so abrupt. Then, in the same paragraph, the author talks about the network output. So mixed, so confused to read.

L106: The present paper created some Multi-head concept? If not, please, put reference . It is not clear in the text.

L110: Softmax is mainly useful on classification which the outputs present parallel probabilities and dependent with max 1. What you mean normalization using softmax since it returns probabilities? In addition, after the softmax, only one output is presented. What is presents indeed? Some unique probability value?  

L129: Replace "very good" for "satisfatory".

L134-135: Please rewrite sentence.."The nonlinearity used 
in the Position-wise Feed-Forward Network block is GELU"

L146: Which sensors? Please cite here.

L148: "The first 11 activities PRESENTED in Table 1..."

L150: "The other 4 activities were recorded outdoors." Was recorded where? Where is outdoor? Which outdoor scenario?

L153: Again. Suddenly the author introduced another information in the same paragraph about different issue. Different contexts must to be in different paragraph.

L156: I recommend to add new paragraph. The subsection 2.3 present only one big paragraph. It is not good.

L165: Change section title to .."Transformer Model for .."

L164: Table header vertical space is to big. I recommend to abbreviating the header names.

L272: Figure 5 needs to be better explained before. What this signals/charts represents? In addition it is so small to see the numbers and values. Please, correct it. Figure 5 is confused!

---------------

Results

---------------

L285: "Figure 8 (ADD ',') ...". Repeat it along all journal when it is at beginning of some sentence. 

L312-313: Please rewrite the sentence, is it confused to read. try to simplify the sentence. 

L319: "THE PROPOSED HAR, ...."

Figure 6, 7, are really bad explained here. Are so many spectrograms and low explanation about!

Figure 8, needs more explanation because it means nothing different between quadrants.

--------------------------------------

Putting all in a nutshell.

--------------------------------------

The present paper presents a not so innovative methodology or method, but it can be accepted by me if, ONLY if, the authors rewrite several parts of the journal (some of them, were cited before).

The present work brings several confused sentences, i.e. in the same sentences and paragraph, we can see  different contexts.

The methodology, seems to have a lot of different methods in a confuse explanation like 'putting all together'. Some method used, e.g., softmax wasn't well explained in the present context, with kind of output it returns (probabilities???)

The citation formats are not usual for MDPI, and the results are sometimes, confused too.

To conclude, due to all these points, I reconsider it after major revision, since abstract til results.

Reviewer 2 Report

You have 20 references. I think I could cite more authors who have worked on this topic

Reviewer 3 Report

The authors present the article entitled “Wearable Sensor Based Human Activity Recognition with Transformer”. However, it is not possible to extend my recommendation for publication according to the next concerns:

The abstract must be rewritten. The abstract section should present a pertinent overview of the proposed work. I suggest reading the author guidelines.

The objective is not clear. I suggest to hard improve the objective by highlighting the novelty of the proposed work.

Avoid citing the works as “papers”. It is possible to use, for example, “Authors from..”, instead

Vectorize the figures in order to see the details.

Figure 1: Please use a different visual effect or colors for a better appreciation

Line 188:  In “m/s2”, please use superindex.

Figure 3: a) or b) are not in the plot. Place where they correspond.

Figures 6 and 7: I recommend synthesizing the information. In the main text, the description of these figures is ambiguous. 

Use the symbol % instead “percent”.

In line “Apart from health monitoring and rehabilitation, it can be used in gaming, human-robot interaction, robotics, or sports [1]. “of the introduction section, it can be re-supported the statement with the following references: Speed controller-based fuzzy logic for a biosignal-feedbacked cycloergometer; Impact of EEG Parameters Detecting Dementia Diseases: A Systematic Review; A high-accuracy mathematical morphology and multilayer perceptron-based approach for melanoma detection

Use spaces when calling a magnitude + unit, “100Hz”

Write a paragraph to introduce the whole section 2 “2. Methods” before 2.1.

This paragraph can be supported by the following references

 “A transformer is a type of neural network, based purely on attention mechanisms, which, like recurrent or convolutional networks, typically process time series and look for correlations between features within time steps. It is frequently used to work with natural language, where it achieves higher scores than recurrent neural networks. The transformer consists of Multi-Head Attention, fully connected, normalization, and dropout layers. It also contains residual connections that help with the gradient backpropagation in a deep neural network.”

A new approach for motor imagery classification based on sorted blind source separation, continuous wavelet transform, and convolutional neural network; Implementation of a socket for hip disarticulation based on ergonomic analysis

My major concern is the novelty of the project since there are lots of similar works. Probably, a comparison table, by considering the findings of the work vs other works, could show the novelty of the project.

Besides, I really like that the authors attempt to simulate and implement the algorithms, but please, show what the project has that others lack.

Please separate the discussion and conclusion into separate sections.

Round 2

Reviewer 3 Report

Thank you for considering my concerns. The manuscript can be accepted.